# Mixing Propolis from Different Apiaries and Harvesting Years: Towards Propolis Standardization?

**DOI:** 10.3390/antibiotics11091181

**Published:** 2022-08-31

**Authors:** Marta Peixoto, Ana Sofia Freitas, Ana Cunha, Rui Oliveira, Cristina Almeida-Aguiar

**Affiliations:** 1Department of Biology, School of Sciences, University of Minho, Campus de Gualtar, 4710-057 Braga, Portugal; 2CITAB, Centre for the Research and Technology of Agro-Environmental and Biological Sciences, University of Minho, 4710-057 Braga, Portugal; 3CBMA, Centre of Molecular and Environmental Biology, University of Minho, 4710-057 Braga, Portugal

**Keywords:** propolis, ethanol extracts, blends of propolis, phenolic compounds, antimicrobial activity, antioxidant potential

## Abstract

Global demand for safe, effective and natural products has been increasing in parallel with consumers’ concerns about personal and environmental health. Propolis, a traditional and potentially medicinal product with several health benefits, is a beehive product with a worldwide reputation. However, despite the bioactivities reported, the low productivity and high chemical heterogeneity have been extensively hampering broader industrial uses. To assist in overcoming some of these problems, we prepared and characterized mixtures of ethanol extracts of a heterogeneous propolis sample (Pereiro) collected over a five-year period (2011–2015) and, additionally, we mixed two different propolis samples from distinct regions of Portugal (Pereiro and Gerês), also harvested at different times. An investigation of the antimicrobial and antioxidant properties, as well as characterization of the chemical composition of the eleven propolis blends were performed in this work. The antioxidant and antimicrobial activities of such blends of propolis samples, either from different localities and/or different years, were maintained, or even enhanced, when a comparison of the individual extracts was conducted. The differences in the chemical composition of the original propolis samples were also diluted in the mixtures. The results reemphasize the great potential of propolis and suggest that mixing different samples, regardless of provenance or harvesting date, can contribute to propolis standardization while simultaneously increasing its availability and adding value to this beehive byproduct.

## 1. Introduction

The global market for natural products has been growing due to increasing consumer concern about personal health and the widespread use of synthetic chemicals with potentially harmful side effects and environmental impacts. The rising need for safe, effective and natural alternatives has given attention to propolis, widely recognized as a traditional and potentially medicinal product with several health benefits [1,2]. Propolis or “bee glue” is a sticky product composed of resinous and balsamic materials collected by bees from several plant sources and also of other substances resulting from bees’ metabolism [3]. Interest in propolis has mostly arisen due to its broad range of valuable bioactivities, such as its antioxidant and antimicrobial properties, which have been mainly ascribed to phenolic compounds [4,5,6].

Worldwide demand for propolis has been rising, with China and India being the fastest-growing markets back in 2015 [7]. Leading the propolis market since 2018, North America is expected to be at the top by 2024 [8]. According to the most recent projections, the global propolis market size is expected to reach around USD 700 million, with a CAGR (Compound Annual Growth Rate) of 6% between 2019 and 2024 [8]. 

Despite the difficulty in accurately estimating the total sales of propolis and its byproducts, as beekeeping is essentially a home-made industry, US propolis sales were estimated at 46,000 € in 1996 [9]. Japan’s interest in propolis has triggered a price increase from 4.4 to 176.4 €/ Kg in recent years, revealing the economic potential of bee glue [10]. However, in other countries, such as Portugal, propolis has still been an undervalued and cheaper beehive product [11], not because of a lower quality but rather due to its low level of exploitation [12,13]. This low level of commercial exploitation by most Portuguese beekeepers is mainly due to their lack of awareness and lack of technical knowledge, along with the rooted tradition of honey production, making propolis production practically neglected in the context of national apiculture [14]. Additionally, propolis productivity is very low: a European hive can produce between 50–150 g propolis per year [15] and, almost similarly, the estimated annual production of Portuguese propolis is around 100 g/ hive [15]. Lower productivities, such as 15.7 g propolis/ hive [16] or 24.2 ± 22.5 g/ hive [17] have been reported worldwide too, contributing to a reduced interest in propolis production as a considerable volume is often required by its main target, the pharmaceutical industry. In addition, until adequate quality parameters are developed, propolis will remain an alternative treatment without acceptance in medicine. The complex and variable propolis chemistry makes propolis quality standardization very challenging. More than 800 different chemical compounds were identified but the link between marker compound(s) for propolis and their respective therapeutic potential is still missing [18].

Some factors have been identified as determinants in propolis production and quality —the resin botanical origin, the genetics of the honey bee, the hive structure and material, food availability, environmental factors, and disease (see [19] for a review)—but the lack of quality standards and proper legislation still hinders the introduction of propolis to the world market. Therefore, a compilation of the parameters that need to be ensured to commercialize propolis for specific applications is critical and regulatory agencies should establish the quality parameters for propolis in a certain country. 

The low productivity of propolis and its great chemical variability allied to the lack of standardization are the main obstacles to propolis applications, for example, in the food industry as a food preservative but especially in the pharmaceutical industry for therapeutic purposes [10,20,21,22]. In this framework, and given the great potential of this natural product [6,23,24], the aim of this work is the evaluation of the antimicrobial and antioxidant activities of mixtures of propolis, as well as the characterization of its chemical composition through in vitro methodologies. We previously showed that mixing ethanol extracts of propolis collected over five years from an apiary (Gerês; G) results in the maintenance or improvement of bioactivities [13]. However, propolis from Gerês seems rather unique, as it shows chemical and biological constancy along harvests in different years [25], which is different to what is largely known for bee glue. As a result, we intended to go further in evaluating (i) the outcome of mixing a very different but more typical propolis sample collected in different years from another apiary (Pereiro; P) and showing different bioactive and chemical profiles over the years [26]; and (ii) the effect of blending the two completely different propolis samples collected from the two apiaries (G and P) in different years.

Thus, in this study we prepared blends of (i) the ethanol extracts of propolis collected in Pereiro over a 5-year period (2011–2015); and of (ii) the ethanol extracts of propolis collected from two distinct apiaries/ regions—Pereiro and Gerês—in selected years. By mixing propolis from different apiaries and harvesting years, we expect to be able to overcome the limitations of the low yield per hive diluting the differences found between individual samples and increasing the available propolis for the market, in this way contributing towards a standardization of propolis biochemical profiles and propolis value.

## 2. Results

### 2.1. Characterization of Mixtures Obtained from Propolis Samples Harvested from the Same Apiary over Different Years

#### 2.1.1. Total Polyphenols and Flavonoids Contents

In order to make a broad chemical characterization of individual extracts and blends of propolis from Pereiro, total polyphenols contents (TPC) and total flavonoid contents (TFC) were determined (Table 1).

The total polyphenols content of mixtures of ethanol extracts of Pereiro propolis (Table 1) range between 194.9 and 215.1 mg GAE/ g propolis extract, displayed by the mixture of P12.EE and P13.EE (mP_(P12.EE+P13.EE)_) and the mixture of all of the five EEs of propolis collected in different years, respectively. The TPC of mP.EEs is frequently amid the contents of the individual EEs of Pereiro propolis. Interestingly, the exception was for the mixtures mP_(P11.EE+P13.EE)_ and mP_(P13.EE+P15.EE)_, with a slightly lower content when compared with the respective individual extracts, which are the ones with the highest TPC. The nature of the method for estimation of the phenolic content might account for such results, as the TPC methodology is based on the reduction of the Folin–Ciocalteu reagent, which could be affected by synergisms and/ or antagonisms of the phenolics that came together when the mixtures were prepared. The method was reported as giving higher values for polyphenols than the sum of the individual compounds as measured by HPLC [27]. Oxidation of multiple phenolic groups may generate products that are themselves reducing agents, thus giving a higher Folin value but the same can potentially occur in vivo and thus the Folin–Ciocalteu measurement may be relevant [28]. However, despite the differences observed between the individual extracts, the TPC of the mixtures were very similar (*p* > 0.05; ^A^). In addition, the TPC of the mP.EEs were similar to the ones found in other single EEs of European propolis samples [29,30], meaning that even when mixing propolis samples with different TPC values, the phenolics contents of the blends remain within the values generally found for propolis from other sources. Moreover, the mixtures obtained from individual extracts with high and low TPC produced blends with intermediate TPC (see the pairings) or high TPC (compare, for instance, mP(_P11.EE − P15.EE)_ and mP_(P11.EE−P13.EE)_ with their respective individual PEEs), highlighting not only that blending diluted the content differences between samples, but also that blending samples of several years did not significantly reduce the high TPC found in the samples of particular years.

The total flavonoids content (Table 1) varied between 43.9 and 67.9 mg QE/ g propolis extract in the mP.EEs, with the higher contents being measured in P15.EE-containing mixtures (from 55.3 to 67.9 mg QE/ g extract), with this individual extract being the one displaying the highest TFC (78.4 mg QE/ g extract). The TFC of the mixtures generally maintained the values of the individual extracts or showed a slight increase. Still, the values were within the range of contents found in other European individual samples [31,32]. In the case of the TFC, the pairings obtained with an extract of high TFC (again those from odd years) always resulted in mixtures retaining this high value or with a significantly higher TFC, but nevertheless within the range of contents found in other European individual samples [31,32]. This increased TFC of the mixtures may be due to the occurrence of some kind of synergism between compounds from different extracts and suggests a higher potential of blends in a broad spectrum of applications. In addition, as shown by Chang et al. [33], this method does not detect flavanones since these compounds do not form stable complexes with Al^3+^. Nevertheless, the contents in flavones and flavonols detected by the assay are reliable indicators of the antioxidant activity of the samples (see further discussion in Figure 1b).

#### 2.1.2. Antioxidant Potential of Propolis

Antioxidant activity, expressed by the EC_50_ parameter, varied between 12.3 and 14.9 µg/mL (Table 2) for Pereiro propolis-containing mixtures. Interestingly, even when the EC_50_ values were very different among the individual extracts of the blend, the antioxidant potential of the majority of mP.EEs was similar to the one of the most-active individual extracts present in the blend (see, for instance, mP_(P11.EE+P12.EE)_ and mP_(P11.EE+P14.EE)_), except for P15.EE-containing blends, which showed an intermediate EC_50_ value. Together, these results highlight the advantage of mixing multiple propolis samples: higher homogeneity between blends/lots, thereby contributing to its standardization, and maintenance of the higher phenolic and flavonoid contents and antioxidant capacity, meaning an improved, more constant and predictable quality of the blends. 

In general, all of the mixtures displayed a similar antioxidant potential (Table 2), once again in the range reported both for single European (26.45 μg/mL Ireland and 27.72 μg /mL Czech Republic) [29,34] and for other individual Portuguese propolis samples [35], even being propolis diverse in this activity as well [36,37]. A lower DPPH• scavenging activity (IC_50_ = 32.35 ± 2.84 μg mL^−1^) was reported for an EE of Chinese propolis [38], whereas Brazilian green propolis appears to have the best antioxidant potential [39].

Figure 1 shows a negative correlation between both the TPC (Figure 1a) and TFC (Figure 1b) values (Table 1) of the P.EEs and their antioxidant capacity, measured by the radical scavenging activity (Table 2); the higher the former, the higher the latter. In addition, it shows that, when mixing propolis individual extracts, the TPC, TFC and EC_50_ values of the mixtures narrowed down its range, clustering near the most active and phenolic-rich individual extracts. Yet, when mixing propolis, the TPC, TFC and radical scavenging activity of the mixtures stood between the values of the individual samples or similar to the most active individual propolis extract. In fact, mP.EEs displayed roughly the same EC_50_ value (Figure 1a,b; black dots) regardless of the TPC (Figure 1a) or the TFC (Figure 1b) of the individual extracts (red dots).

These observations suggest that mixing different known propolis samples will ensure that the bioactivities of the mixtures will be in the range of the individual samples, making the choice of blends more predictable in terms of their bioactivity, thereby contributing to normalizing the antioxidant potential.

#### 2.1.3. Antimicrobial Potential of Propolis

Antibacterial properties of EEs and of mixtures of EEs of propolis from Pereiro samples are presented in Table 3. In most cases mP.EEs showed antibacterial activity that was identical to the most-active single extract, as seen in mP_(P11.EE−P13.EE)_ against *Bacillus subtilis* and *Staphylococcus aureus*, or an even higher activity such as in mP_(P11.EE+P12.EE)_ against *B. subtilis* or mP_(P13.EE+P15.EE)_ against *S. aureus*. 

The same type of result was observed against more-resistant bacteria, such as methicillin-resistant *Staphylococcus aureus* (MRSA) or *Escherichia coli*, with mP_(P12.EE+P13.EE)_ and mP_(P13.EE+P15.EE)_ as the most active mixtures (MIC = 1000 µg/mL). mP.EEs displayed a higher antibacterial activity against *E. coli* than individual Portuguese propolis samples studied by [40], though were similar in their activity against *S. aureus*. The remaining mixtures showed a similar antibacterial activity, displaying identical MIC values and being more active against Gram-positive than towards Gram-negative bacteria, as generally reported [21]. Additionally, MIC values of propolis mixtures against *S. aureus*, *B. subtilis* and *E. coli* were very close to the average MIC calculated by [41] from several studies (mean MIC values of EEs of propolis were 457, 180 and 784 µg/mL respectively).

According to Saraiva et al. [42], plant extracts with MIC ranging from 100 to 500 µg/mL are active extracts and moderately active if MIC varies from 500 to 1000 µg/mL while extracts with MIC ranging from 1000 to 2000 µg/mL are considered to have low activity. Transposing this classification to propolis extracts, mP.EEs were revealed to have particularly interesting activities against *B. subtilis*, *Propionibacterium acnes* and *S. aureus*. Moreover, although no data were obtained for the individual extracts against *P. acnes,* for instance, the MIC of the mixtures and the findings regarding its behavior against the other indicator strains support the choice of blends over individual extracts and open perspectives for clinical applications in the case of this acne-causing strain.

The results obtained for mP.EEs anti-yeast activity followed the same pattern as it did for antibacterial activity. The mixtures mostly displayed an MIC value similar to the most-active extract or, in some blends, even lower MIC values (Table 4). For *S. cerevisiae*, for example, almost all of the mixtures displayed the MIC of the most-active extract, except mP_(P11.EE+P13.EE)_, which was even more active than the single extracts.

A curious observation from all of these results is that P14.EE, one of the extracts with the lowest TPC and TFC (Table 1) and antioxidant capacity (Table 2) was the one with the highest antibacterial activity (Table 3) but did not have a high anti-yeast activity (Table 4), where typically P15.EE and P15.EE-mixtures excel. This seems to suggest that propolis mode of action against bacteria is not as tightly related to its level of polyphenols and flavonoids contents per se, but possibly is related to certain combinations of compounds or specific synergisms, as has been reported [43].

### 2.2. Characterization of Mixtures of Propolis Obtained from Two Apiaries and Different Harvesting Years

It is well known that propolis composition depends on a myriad of factors and that chemical and biological diversity are propolis signatures [21,22,44]. Over the years, propolis from Gerês, in the north of Portugal, has shown great consistency in terms of its biological activity and phenolic profiles [25]. On the contrary, propolis from Pereiro, in the center of the country, has shown great heterogeneity over the years [26]. According to our findings in this study, blending may potentiate propolis bioactivities and somehow allow us to standardize its characteristics by diluting year-dependent variability. Similar findings were previously reported for propolis from Gerês [13], which has been showing remarkably constant chemical and biological profiles [25]. This constant behaviour may be explained by the surrounding vegetation of the Gerês apiary, which belongs to a protected area of a National Park, by the apiculture practices (organic beekeeping) as well as the standardized way of propolis production and harvesting (from grids, rather than being harvested by scraping during the annual cleaning of the beehive, as in Pereiro). Therefore, we hypothesized that mixing samples from different apiaries and harvesting years could result in a similar outcome in terms of potency. Hence, we investigated the bioactivities of blends based on propolis samples from Gerês and Pereiro.

#### 2.2.1. Total Polyphenols and Flavonoids Contents

When EEs of propolis from the two apiaries—Gerês and Pereiro—were mixed in two different blends, the TPC of these mixtures mP + G were similar (Table 5) and comparable to the TPC values previously observed for mP.EEs (Table 1) (*p* > 0.05), though it seemed that G.EE was superimposed in the mixture. Thus, mixing propolis with very different characteristics, collected from different apiaries and regions of the country as well as from different harvesting years does not appear to alter the range of the TPC found in mixtures of propolis from the same apiary (Table 1). In addition, as previously observed for blends of propolis from Gerês and from different years [13], the TPC values became closer, with no significant differences between the mixtures, suggesting that mixing propolis samples can contribute to standardization.

On the contrary, the two tested mP + G mixtures showed different TFC values, the highest was found in the P15.EE-containing mixture (Table 5), yet an intermediate value between the ones of the respective individual extracts. Osés et al. [45] also found a high but diverse TFC in 13 samples of propolis from different American and European regions, with values ranging between 18.48 and 83.76 mg QE/ g extract, notwithstanding that all had a strong antioxidant activity, despite their TFC variation.

#### 2.2.2. Antioxidant Potential

In the blends made with EEs of propolis from two different localities, there was also an improvement of antioxidant capacity relative to the individual extracts, particularly in m_(P14.EE+G15.EE)_ (Table 6). Despite the EC_50_ differences between the individual extracts, both mixtures showed a similar antioxidant potential (Table 6), once again suggesting that mixing propolis may contribute to the standardization of this natural product.

Thus, the use of mixtures appears to be beneficial, allowing us to make use of even the least-active propolis samples without a loss of bioactivity or even their potentiation. Mixing propolis, whether from a single apiary and different harvesting years, or from different apiaries and harvesting years, can contribute to the standardization of bioactivity, as different mixtures show very close EC_50_, regardless of either TPC or TFC values or the EC_50_ values of the individual samples (Figure 2), although in these mixtures the best correlation with antioxidant capacity was found in the TFC.

#### 2.2.3. Antimicrobial Potential of Propolis Blends

Propolis antibacterial activity can occur directly against microorganisms or indirectly by stimulating the immune system [41,46]. Mixtures made with propolis from Pereiro and Gerês showed MIC values that were generally different to the ones displayed by the individual extracts (Table 7), albeit no loss of activity was detected. For *B. subtilis,* for example, the mixtures were found to have the lowest MIC or an intermediate value between the two individual extracts. Al-Waili [47] recently reported that mixtures of propolis from two different regions of Iraq displayed greater antimicrobial activity and faster wound-healing than individual propolis samples, also showing a greater potential of the mixtures compared with the individual propolis extracts.

Regarding their anti-yeast activity (Table 8), the mixtures mP + G are less active than P14.EE and P15.EE, but more active (or as active as) the EEs of propolis from Gerês. Although antibacterial activity is the most prominent antimicrobial propolis property, propolis mixtures can display antifungal activity to the same extent as individual propolis samples against *C. albicans*, for instance. Antifungal activity has been correlated with several propolis constituents, such as chrysin and cinnamic acid derivatives, and this knowledge can contribute to the selection of molecules or propolis samples with higher activity and effectiveness in antifungal treatments [48].

The antimicrobial activity either of extracts or of mixtures of propolis falls within the variability described for this bioactivity in European propolis [25,26,38,41,44]. More importantly, our results indicate that mixing propolis extracts—either from different regions of the country and/ or from different harvesting years—does not promote the loss of antimicrobial activity. Instead, this activity can even be improved in some cases [13]. Furthermore, all of the ethanol extracts used in this study were prepared in the years that the propolis samples were collected and did not lose either antioxidant or antimicrobial activities when used some years later (not shown). Evidence has also been given that propolis in an ethanol solution over a period of 10–15 years can increase its antibacterial activity [49] and that samples of fresh and aged propolis have similar qualitative composition, radical scavenging and antimicrobial properties [50]. In a recent work, we suggested that aged propolis should not be discarded but explored for alternative applications, as a propolis leftover (with more than a year of storage) showed antimicrobial activity [51]. Furthermore, although collected and prepared over four different years (2011–2014), G.EEs chemical profiles showed a huge similarity regarding the type of phenolic compounds, with variations being mostly quantitative [25]. This maintenance of propolis properties for long periods is another asset in propolis conservation and the production of propolis mixtures from samples that are stored for a long time.

## 3. Materials and Methods

### 3.1. Propolis Origin and Ethanol Extraction

Propolis samples were collected over a period of five years (between 2011 and 2015), in apiaries from two different regions of Portugal. One of the apiaries is located in Guarda district, in Pereiro (40°44′57.135″ N; 7°0′59.403″ O) and the other is in Gerês (41°45′41.62″ N; 7°58′03.34″ W). Samples were collected annually between August and September and were identified by the letters P or G, according to their origin (Pereiro and Gerês, respectively) followed by two digits corresponding to the harvest year (for example, P12 corresponds to propolis of Pereiro collected in 2012). Sample P was obtained by scraping whereas G was collected from grids.

All propolis samples were extracted with ethanol in the same year of collection, following the methodology reported [25]. Briefly, 15 g of propolis was incubated for 24 h with 80 mL of absolute ethanol, in the dark at room temperature (RT) at 110 revolutions per minute (rpm). The suspension was filtered and the resulting residue was further extracted as described above but with 50 mL of absolute ethanol. Filtrates resulting from the two-step extraction process were pooled and the solvent was evaporated on a rotavapor (Buchi, Flawil, Switzerland) connected to a bath at 38–40 °C and at 47 rpm. The ethanol extracts (EE) prepared with propolis from Pereiro (P.EEs)—P11.EE, P12.EE, P13.EE, P14.EE and P15.EE—and from Gerês (G.EEs)—G11.EE, G12.EE, G13.EE, G14.EE and G15.EE—were stored in the dark at 4 °C, until needed.

### 3.2. Preparation of Blends of Propolis Ethanol Extracts

#### 3.2.1. Mixtures of Ethanol Extracts of Propolis from Pereiro

Mixtures of EEs of propolis from the apiary Pereiro (mP.EE) were prepared considering the antioxidant, antibacterial and antifungal activities of each individual extract. Firstly, each EE was classified into three categories labeled as: “most active”, “least active” and “intermediate” (the remaining extracts) for each of the bioactivities, similar to that which was previously conducted for propolis from Gerês [13]. Based on this categorization, mP.EEs were prepared according to the following criteria: (i) the most and the least active extracts, (ii) an intermediate and the least active, (iii) an intermediate and the most active, (iv) all the intermediate P.EEs and (v) a mixture of all of the P.EEs. Thirteen mixtures were planned according to these five criteria but only nine different mixtures were prepared, as some of the blends shared the same composition (Table 9). 

#### 3.2.2. Mixtures of Ethanol Extracts of Propolis from Pereiro and from Gerês

Two blends were prepared with propolis from the two apiaries. For this purpose, P.EEs and G.EEs were used, taking into account the most and least active extracts in terms of antioxidant capacity (Table 10).

All blends were prepared from P.EEs (Table 9) or from P.EEs and G.EEs (Table 10) at the same concentration, 10 mg/ mL, and adding equal volumes of each individual extract of the blend. The mixtures were then used for chemical characterization and biological activity assays.

### 3.3. Determination of Total Poliphenols Contents

The total polyphenols content (TPC) was determined following an adaptation of the Folin–Ciocalteu colorimetric method [5,52]. A volume of 50 µL of propolis ethanol extracts or mixtures—prepared in a concentration range of 10 to 200 µg/mL in absolute ethanol—were added to 50 µL of 10% (*v*/*v*) Folin–Ciocalteu reagent and 40 µL of 7.5% (*w*/*v*) Na_2_CO_3_. After 1 h incubation in the dark at RT, the absorbance was measured at 760 nm. The results are expressed in milligrams of gallic acid equivalents (GAE) per gram of propolis extract (mg GAE/ g extract), upon a calibration curve performed with gallic acid at concentrations between 5 and 50 µg/mL.

### 3.4. Determination of Total Flavonoids Contents

The total flavonoids content (TFC) was determined following the Woisky and Salatino method [53]. Propolis EEs or mixtures were prepared in the concentrations of 100 and 1400 µg/mL in absolute ethanol and mixed (50 µL) with 50 µL of 2% (*w*/*v*) AlCl_3_. The absorbance was measured at 420 nm after 1 h incubation in the dark at RT. Quercetin varying from 5 to 200 μg/ ml was used as a standard. The results are presented as milligrams of quercetin equivalents (QE) per gram of propolis extract (mg QE/ g extract).

### 3.5. Determination of of DPPH• Scavenging Potential

The radical *2,2-diphenyl-1-picrylhydrazyl* (DPPH•) is a stable free radical which is reduced in the presence of hydrogen-donating antioxidants, promoting a color change that can be measured by spectrophotometry [54]. The DPPH• scavenging activity of propolis was determined by mixing 50 μL of samples at concentrations of 1 to 50 μg/mL (in absolute ethanol) and 100 μL of 0.004% (*w*/*v*) DPPH•, followed by 20 min incubation at RT, in the dark, and an absorbance measurement at 517 nm. The percentage reduction for each tested concentration was calculated using the following equation:(1)$$Inhibition\;\left(\%\right)=\times 100$$
where A_Sample_ is the absorbance of the extract with DPPH• and A_Control_ the absorbance of the control (DPPH• and ethanol). EC_50_ (µg/mL), which defines the propolis concentration needed to scavenge 50% of the free radical, was calculated by interpolation from those values. Gallic acid was used as a standard.

### 3.6. Evaluation of Antimicrobial Activity 

To evaluate propolis antimicrobial activity, a panel of microorganisms was selected, taking into consideration their susceptibility and their clinical and pharmaceutical importance. Gram-negative (*Escherichia coli* CECT 423) and Gram-positive bacteria (*Bacillus subtilis* 48886, *Bacillus cereus* ATCC 7064, *Bacillus megaterium*, *Propionibacterium acnes* H60803, *Staphylococcus aureus* ATCC 6538 and *Methicillin-resistant Staphylococcus aureus* M746665 (MRSA)), as well as the yeast *Candida albicans* 53B and *Saccharomyces cerevisiae* BY4741, were all provided by the culture collection of the Department of Biology of the University of Minho. Bacteria were cultured in LB medium (Difco) for 24 h at 37 °C and 200 rpm and yeast in YPD medium (Difco) for 48 h at 30 °C and 200 rpm. Agar 2% (*w*/*v*) was added to each recipe to prepare solid media (LBA and YPDA).

An adaptation of the agar dilution method was used to determine the antibacterial and antifungal activities of all of the propolis mixtures. Each mixture was incorporated in LBA and YPDA media at various concentrations (25, 50, 100, 200, 500, 750, 1000, 1500 or 2000 μg/ ml) depending on the strain under study. Subsequently, 5 µL drops of exponential phase microbial cultures (OD_600_ = 0.4–0.6) were transferred to the Petri dishes, in triplicate, with incubation for 24 h at 37 ºC in the case of bacteria, or 48 h at 30 ºC in the case of yeast. Minimum inhibitory concentration (MIC) values were obtained upon observation of the presence/absence of microbial growth. Experiments were repeated three times.

### 3.7. Statistical Analysis

All of the assays for chemical characterization and antioxidant potential were analyzed and the results were presented as mean ± standard deviation from a variable number of assays, always equal to or greater than three (*n* ≥ 3). GraphPad Prism for Windows (version 8.0.1, GraphPad Software, San Diego, California USA, www.graphpad.com, accessed on 2 December 2021 was used in the statistical analysis of the results. The results were analyzed using analysis of variance (ANOVA) followed by the Tukey test. Differences considered statistically significant (*p* ≤ 0.05) were expressed with the alphabetical notation system, using different letters (lowercase when comparing mixtures and individual extracts, uppercase when comparing mixtures).

## 4. Conclusions

This study is a first approach to mixing propolis samples from different years and regions, and consequently with different bio and chemical profiles. We found that when mixing propolis from the same apiary but collected over different years, or when combining propolis samples collected from different regions and years, the antimicrobial and antioxidant activities of the most active of the individual extracts were either preserved or enhanced. Considering the results regarding the chemical characterization (TPC and TFC) and bioactivities (antioxidant and antimicrobial activities) of the mixtures, we can also conclude that the differences between individual propolis samples can be attenuated and a reduction in heterogeneity was obtained, thereby contributing to propolis standardization. These findings support the great potential of propolis and add even more value to this hive resource. Such valorization is also related to a greater use of the product, since samples from different years can be used without any loss of bioactivity. This efficiency in combining different propolis extracts/ samples can contribute to increasing beekeepers’ interest in this product and enable them to face larger demands for this natural product [55]. Together, these main outcomes are two important starting points for the valorization and standardization of propolis.

Characterization and quantification of propolis bioactive molecules, such as polyphenols, work as a fingerprint of propolis samples, being of interest in medicine and nutraceuticals [56] and several other applications. With the possible standardization of propolis, its application in combating several health problems, such as obesity and associated metabolic disorders becomes easier [57], or as an antioxidant and anti-inflammatory agent in the prevention and care of various diseases [58,59].

## Figures and Tables

**Figure 1 antibiotics-11-01181-f001:**
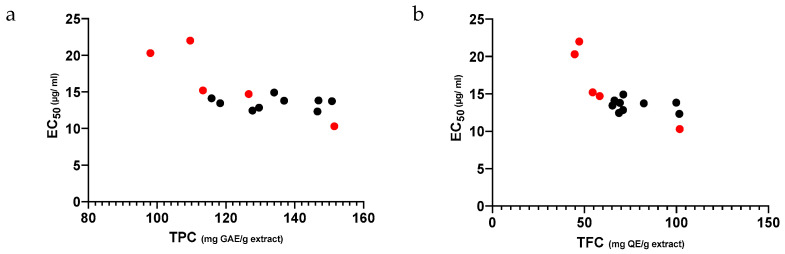
Correlation between antioxidant potential (EC_50_ values)) and TPC (**a**) or TFC (**b**) of individual ethanol extracts of propolis from Pereiro (●) and of mP.EEs (●).

**Figure 2 antibiotics-11-01181-f002:**
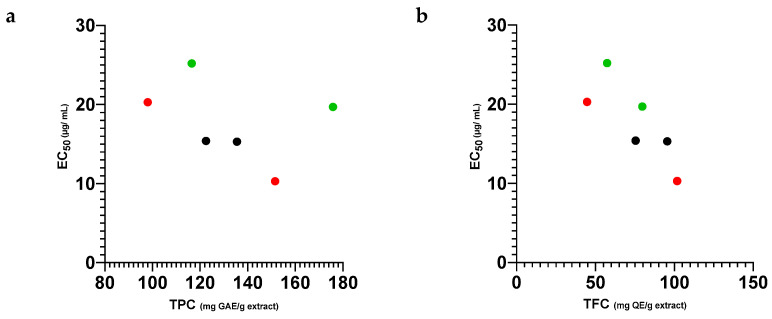
Correlation between antioxidant potential (EC_50_ values) and TPC (**a**) and TFC (**b**) of individual ethanol extracts of propolis from Pereiro (●) and from Gerês (●) and of propolis blends (mP + G) (●).

**Table 1 antibiotics-11-01181-t001:** Total polyphenols (TPC) and flavonoids contents (TFC) of the ethanol extracts of propolis harvested at Pereiro in the years 2011 to 2015 and of the mixtures of P.EEs prepared in this work. The results are presented as mean ± standard deviation of mg of gallic acid equivalents (GAE) and mg of quercetin equivalents (QE) per g extract (mg GAE/ g extract and mg QE/ g extract per g of extract), respectively. Significant differences (*p* < 0.05) between TPC or between TFC of single extracts (P.EEs) and mixtures of extracts (mP.EEs) are noted with different lowercase letters. Significant differences between TPC or between TFC of mP.EEs (*p* < 0.05) are represented with different uppercase letters.

Samples	TPC (mg GAE/ g Extract)	TFC (mg QE/ g Extract)
P11.EE	224.6 ± 11.5^a^	43.4 ± 0.7^a^
P12.EE	173.2 ± 10.1^c^	32.4 ± 1.5^b^
**mP_(P11.EE+P12.EE)_**	**206.6 ± 4.5^b; A^**	**43.9 ± 2.0^a; C^**
P11.EE	224.6 ± 11.5^a^	43.4 ± 0.7^a^
P13.EE	217.6 ± 6.6^a^	38.1 ± 2.0^b^
**mP_(P11.EE+P13.EE)_**	**200.3 ± 6.3^b; A^**	**46.2 ± 0.8^a; C^**
P11.EE	224.6 ± 11.5^a^	43.4 ± 0.7^b^
P14.EE	174.4 ± 4.2^c^	33.4 ± 2.0^c^
**mP_(P11.EE+P14.EE)_**	**198.5 ± 9.3^b; A^**	**47.7 ± 0.5^a; C^**
P12.EE	173.2 ± 10.1^b^	32.4 ± 1.5^c^
P13.EE	217.6 ± 6.6^a^	38.1 ± 2.0^b^
**mP_(P12.EE+P13.EE)_**	**194.9 ± 12.2^b; A^**	**44.6 ± 2.7^a; C^**
P13.EE	217.6 ± 6.6^a^	38.1 ± 2.0^b^
P14.EE	174.4 ± 4.2^b^	33.4 ± 2.0^b^
**mP_(P13.EE+P14.EE)_**	**202.5 ± 10.5^a; A^**	**47.8 ± 2.1^a; C^**
P13.EE	217.6 ± 6.6^b^	38.1 ± 2.0^c^
P15.EE	262.2 ± 4.3^a^	78.4 ± 1.7^a^
**mP_(P13.EE+P15.EE)_**	**204.6 ± 15.5^b; A^**	**67.9 ± 2.0^b; A^**
P14.EE	174.4 ± 4.2^c^	33.4 ± 2.0^c^
P15.EE	262.2 ± 4.3^a^	78.4 ± 1.7^a^
**mP_(P14.EE+P15.EE)_**	**209.2 ± 16.5^b; A^**	**67.0 ± 2.2^b; A^**
P11.EE	224.6 ± 11.5^a^	43.4 ± 0.7^a^
P12.EE	173.2 ± 10.1^b^	32.4 ± 1.5^c^
P13.EE	217.6 ± 6.6^a^	38.1 ± 2.0^b^
**mP_(P11.EE−P13.EE)_**	**202.6 ± 12.4^a; A^**	**46.5 ± 2.6^a; C^**
P11.EE	224.6 ± 11.5^b^	43.4 ± 0.7^c^
P12.EE	173.2 ± 10.1^c^	32.4 ± 1.5^e^
P13.EE	217.6 ± 6.6^b^	38.1 ± 2.0^d^
P14.EE	174.4 ± 4.2^c^	33.4 ± 2.0^e^
P15.EE	262.2 ± 4.3^a^	78.4 ± 1.7^a^
**mP_(P11.EE−P15.EE)_**	**215.1 ± 11.3^b; A^**	**55.3 ± 1.7^b; B^**

Note: mP—mixtures of P.EEs. Different lowercase letters (a, b, c, d, e) were used for significant differences (*p* < 0.05) between TPC or between TFC of single extracts (P.EEs) and mixtures of extracts (mP.EEs). Different uppercase letters (**A**, **B**, **C**) were used for significant differences between TPC or between TFC of mP.EEs (*p* < 0.05).

**Table 2 antibiotics-11-01181-t002:** P.EEs’ and mP.EEs’ ability to capture DPPH• free radicals. Antioxidant potential is expressed as a mean ± standard deviation of EC_50_ values (µg/mL). Significant differences between single P.EEs and mP.EEs (*p* < 0.05) are noted with different lowercase letters. Significant differences between mP.EEs (*p* < 0.05) are represented with different uppercase letters.

Samples	EC_50_ (µg/mL)
P11.EE	14.7 ± 2.7^a^
P12.EE	22.0 ± 0.4^b^
**mP_(P11.EE+P12.EE)_**	**13.5 ± 0.3^a; A,B,C^**
P11.EE	14.7 ± 2.7^a^
P13.EE	15.2 ± 2.3^a^
**mP_(P11.EE+P13.EE)_**	**12.5 ± 0.2^a; A,B,C^**
P11.EE	14.7 ± 2.7^b^
P14.EE	20.3 ± 0.3^a^
**mP_(P11.EE+P14.EE)_**	**12.8 ± 0.3^b; A,B,C^**
P12.EE	22.0 ± 0.4^a^
P13.EE	15.2 ± 2.3^b^
**mP_(P12.EE+P13.EE)_**	**14.1 ± 0.7^b; A,B^**
P13.EE	15.2 ± 2.3^b^
P14.EE	20.3 ± 0.3^a^
**mP_(P13.EE+P14.EE)_**	**14.9 ± 0.4^b; A^**
P13.EE	15.2 ± 2.3^a^
P15.EE	10.3 ± 1.7^b^
**mP_(P13.EE+P15.EE)_**	**12.3 ± 0.2^a,b; C^**
P14.EE	20.3 ± 0.3^a^
P15.EE	10.3 ± 1.7^c^
**mP_(P14.EE+P15.EE)_**	**13.8 ± 0.6^b; A,B^**
P11.EE	14.7 ± 2.7^b^
P12.EE	22.0 ± 0.4^a^
P13.EE	15.2 ± 2.3^b^
**mP_(P11.EE−P13.EE)_**	**13.8 ± 0.7^b; A,B^**
P11.EE	14.7 ± 2.7^c^
P12.EE	22.0 ± 0.4^a^
P13.EE	15.2 ± 2.3^c,b^
P14.EE	20.3 ± 0.3^a,b^
P15.EE	10.3 ± 1.7^d^
**mP_(P11.EE−P15.EE)_**	**13.7 ± 0.4^c; B^**

Note: mP—mixtures of P.EEs. Different lowercase letters (a, b, c, d) were used for significant differences (*p* < 0.05) between TPC or between TFC of single extracts (P.EEs) and mixtures of extracts (mP.EEs). Different uppercase letters (**A**, **B**, **C**) were used for significant differences between TPC or between TFC of mP.EEs (*p* < 0.05).

**Table 3 antibiotics-11-01181-t003:** MIC values (µg/mL) of the ethanol extracts of propolis from Pereiro collected from 2011 to 2015 and of the mixtures of P.EEs against *Bacillus subtilis*, *Propionibacterium acnes*, *Staphylococcus aureus*, methicillin-resistant *Staphylococcus aureus* (MRSA) and *Escherichia coli*. The results show the mean ± standard deviation of three assays with three replicates each.

	Gram-Positive				Gram-Negative
Samples	*B. subtilis*	*P. acnes*	*S. aureus*	MRSA	*E. coli*
P11.EE	500	-----	>750	-----	-----
P12.EE	500	-----	500	-----	>1000
**mP_(P11.EE+P12.EE)_**	**200**	**500**	**500**	**1500**	**1500**
P11.EE	500	-----	>750	-----	-----
P13.EE	200	-----	750	-----	-----
**mP_(P11.EE+P13.EE)_**	**200**	**500**	**750**	**1500**	**1500**
P11.EE	500	-----	>750	-----	-----
P14.EE	100	-----	500	-----	-----
**mP_(P11.EE+P14.EE)_**	**200**	**500**	**500**	**1500**	**1500**
P12.EE	500	-----	500	-----	>1000
P13.EE	200	-----	750	-----	-----
**mP_(P12.EE+P13.EE)_**	**200**	**500**	**500**	**1000**	**1000**
P13.EE	200	-----	750	-----	-----
P14.EE	100	-----	500	-----	-----
**mP_(P13.EE+P14.EE)_**	**200**	**500**	**500**	**1500**	**1500**
P13.EE	200	-----	750	-----	-----
P15.EE	500	200	750	>1250	>1250
**mP_(P13.EE+P15.EE)_**	**200**	**500**	**500**	**1000**	**1000**
P14.EE	100	-----	500	-----	-----
P15.EE	500	200	750	>1250	>1250
**mP_(P14.EE+P15.EE)_**	**200**	**500**	**500**	**>1500**	**>1500**
P11.EE	500	-----	>750	-----	-----
P12.EE	500	-----	500	-----	>1000
P13.EE	200	-----	750	-----	-----
**mP_(P11.EE−P13.EE)_**	**200**	**500**	**500**	**1500**	**1500**
P11.EE	500	-----	>750	-----	-----
P12.EE	500	-----	500	-----	>1000
P13.EE	200	-----	750	-----	-----
P14.EE	100	-----	500	-----	-----
P15.EE	500	200	750	>1250	>1250
**mP_(P11.EE−P15.EE)_**	**200**	**500**	**500**	**1500**	**1500**

Note: mP—mixtures of P.EEs.

**Table 4 antibiotics-11-01181-t004:** MIC values (µg/mL) of the ethanol extracts of propolis from Pereiro collected from 2011 to 2015 and of the mixtures of P.EEs against *Saccharomyces cerevisiae* and *Candida albicans*. The results show the mean ± standard deviation of three assays with three replicates each.

Samples	*S. cerevisiae*	*C. albicans*
P11.EE	750	500
P12.EE	>1000	750
**mP_(P11.EE+P12.EE)_**	**750**	**750**
P11.EE	750	500
P13.EE	750	750
**mP_(P11.EE+P13.EE)_**	**500**	**750**
P11.EE	750	500
P14.EE	750	>750
**mP_(P11.EE+P14.EE)_**	**750**	**750**
P12.EE	>1000	750
P13.EE	750	750
**mP_(P12.EE+P13.EE)_**	**750**	**750**
P13.EE	750	750
P14.EE	750	>750
**mP_(P13.EE+P14.EE)_**	**750**	**750**
P13.EE	750	750
P15.EE	500	500
**mP_(P13.EE+P15.EE)_**	**500**	**500**
P14.EE	750	>750
P15.EE	500	500
**mP_(P14.EE+P15.EE)_**	**500**	**750**
P11.EE	750	500
P12.EE	>1000	750
P13.EE	750	750
**mP_(P11.EE−P13.EE)_**	**750**	**750**
P11.EE	750	500
P12.EE	>1000	750
P13.EE	750	750
P14.EE	750	>750
P15.EE	500	500
**mP_(P11.EE−P15.EE)_**	**500**	**750**

Note: mP—mixtures of P.EEs.

**Table 5 antibiotics-11-01181-t005:** Total polyphenols and flavonoids contents of the mixtures prepared with ethanol extracts of propolis from Pereiro and from Gerês, considering the most and the least active extract with regard to its antioxidant activity. The TPC and TFC of each EE are included for ease of analysis. The results are presented as mean ± standard deviation of mg of gallic acid equivalents per g of extract (mg GAE/ g extract) and mg of quercetin equivalents per g extract (mg QE/ g extract) for the TPC and TFC, respectively. Significant differences between single EEs (P and G) and mixtures (mP + G) (*p* < 0.05) are noted with different lowercase letters. Significant differences (*p* < 0.05) between mP + G are represented with different uppercase letters.

Samples	TPC (mg GAE/ g Extract)	TFC (mg QE/ g Extract)
P14.EE	174.4 ± 4.2^b^	33.4 ± 2.0^b^
G15.EE	207.9 ± 7.5^a^	51.7 ± 0.9^a^
**mP + G_(P14.EE+G15.EE)_**	**207.2 ± 6.8^a; A^**	**38.5 ± 3.7^b; B^**
P15.EE	262.2 ± 4.3^a^	78.4 ± 1.7^a^
G13.EE	205.8 ± 3.5^b^	32.6 ± 0.8^c^
**mP + G_(P15.EE+G13.EE)_**	**217.1 ± 9.1^b; A^**	**53.7 ± 4.1^b; A^**

Note: mP + G—mixtures of a P.EE and a G.EE. Different lowercase letters (a, b, c) were used for significant differences (*p* < 0.05) between TPC or between TFC of single extracts (P.EEs) and mixtures of extracts (mP.EEs). Different uppercase letters (**A**, **B**) were used for significant differences between TPC or between TFC of mP.EEs (*p* < 0.05).

**Table 6 antibiotics-11-01181-t006:** The ability to capture DPPH• free radicals of the P.EE and G.EE mixtures is expressed as a mean ± standard deviation of EC_50_ values (µg/ mL). Significant differences between single EEs (P and G) and mixtures (mP + G) (*p* < 0.05) are noted with different lowercase letters. Significant differences between mP + G (*p* < 0.05) are represented with different uppercase letters.

Samples	EC_50_ (µg/ mL)
P14.EE	20.3 ± 0.3^a^
G15.EE	19.7 ± 8.8^a^
**mP + G_(P14.EE+G15.EE)_**	**15.4 ± 1.5^b; A^**
P15.EE	10.3 ± 1.7^c^
G13.EE	25.2 ± 2.5^a^
**mP + G_(P15.EE+G13.EE)_**	**15.3 ± 2.0^b; A^**

Note: mP + G—mixtures of a P.EE and a G.EE. Different lowercase letters (a, b, c) were used for significant differences (*p* < 0.05) between TPC or between TFC of single extracts (P.EEs) and mixtures of extracts (mP.EEs). Different uppercase letters (**A**) were used for significant differences between TPC or between TFC of mP.EEs (*p* < 0.05).

**Table 7 antibiotics-11-01181-t007:** MIC values (µg/ ml) of P.EEs and G.EEs blends against *Bacillus subtilis*, *Propionibacterium acnes*, *Staphylococcus aureus*, methicillin-resistant *Staphylococcus aureus* (MRSA) and *Escherichia coli*. The results show the mean ± standard deviation of three assays with three replicates each.

	Gram-Positive				Gram-Negative
Samples	*B. subtilis*	*P. acnes*	*S. aureus*	MRSA	*E. coli*
P14.EE	100	-----	500	-----	-----
G15.EE	50	50	>750	>1250	>1250
**mP + G_(P14.EE+G15.EE)_**	**50**	**500**	**500**	**1000**	**1000**
P15.EE	500	200	750	>1250	>1250
G13.EE	50	-----	200	>2000	>2000
**mP + G_(P15.EE+G13.EE)_**	**200**	**500**	**750**	**1000**	**1500**

Note: mP + G—mixtures of a P.EE and a G.EE.

**Table 8 antibiotics-11-01181-t008:** MIC values (µg/mL) of P.EEs and G.EEs mixtures against *Saccharomyces cerevisiae* and *Candida albicans*. The results show the mean ± standard deviation of three assays with three replicates each.

Samples	*S. cerevisiae*	*C. albicans*
P14.EE	750	>750
G15.EE	>1500	1000
**mP + G_(P14.EE+G15.EE)_**	**1500**	**1000**
P15.EE	500	500
G13.EE	>2000	>2000
**mP + G_(P15.EE+G13.EE)_**	**1500**	**1500**

Note: mP + G—mixtures of a P.EE and a G.EE.

**Table 9 antibiotics-11-01181-t009:** Composition of blends of P.EEs based on five criteria and taking into account the classification of each P.EE regarding antioxidant, antifungal and antibacterial activities.

Criteria used in P.EEs Mixtures	Antioxidant Activity	Antifungal Activity	Antibacterial Activity
Most active + least active	mP_(P15.EE + P14.EE)_	mP_(P11.EE+P12.EE)_	mP_(P14.EE+P11.EE)_
Intermediate + least active	mP_(P13.EE+P14.EE)_^β^	mP_(P13.EE+P12.EE)_^γ^	mP_(P13.EE+P11.EE)_^α^
Most active + intermediate	mP_(P13.EE+P15.EE)_	mP_(P13.EE+P11.EE)_^α^	mP_(P13.EE+P14.EE)_^β^
Mixture of intermediates	mP_(P11.EE−P13.EE)_	mP_(P13.EE+P14.EE)_^β^	mP_(P13.EE+P12.EE)_ ^γ^
Mixture of all the extracts	mP_(P11.EE−P15.EE)_		

Note: mP—mixtures of P.EEs; β, α, γ—mixtures with the same formulation.

**Table 10 antibiotics-11-01181-t010:** Mixtures of ethanol extracts of propolis from Pereiro (P) and from Gerês (G) prepared with the most and least active extracts of each apiary regarding antioxidant potential.

Criteria Underlying the Mixtures	Mixtures
Most active (G) + least active (P)	mP + G_(P14.EE+G15.EE)_
Most active (P) + least active (G)	mP + G_(P15.EE+G13.EE)_

Note: mP + G—mixtures of a P.EE and a G.EE.

## Data Availability

Not applicable.

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
