# Peer review of "Mixing Propolis from Different Apiaries and Harvesting Years: Towards Propolis Standardization?"

_antibiotics, 2022, doi:10.3390/antibiotics11091181_

Round 1
Reviewer 1 Report
I found the mixture of regions interesting
The mixture potentiated the biological effects.
Author Response
We are grateful for the reviewer's comment. In fact, all our findings suggest that mixing propolis samples can represent a gain either in terms of biological effects or the profitability of propolis leftovers and residues.Reviewer 2 Report
The standardisation issues with phytochemicals are a well-known challenge and this paper makes an interesting and important contribution to addressing the challenge by using the well-known agent propolis. Hence the relevance of the study is quite high.
A number of minor changes are needed to improve the overall quality of the paper.
1. In the introduction it would be worthwhile pointing out some of the variations in quality and yield of propolis from different countries. The text does discuss productivity and yields but not quality in terms of the desired biological activities, hence it would be useful to include this aspect. It also provides greater context in terms of any claim that a particular location is a source for propolis that is "rather unique". This aspect should be explained more - making reference to factors that might explain such stable chemical composition (e.g. less climate change than in other regions over the same period of time). This same point arises in the text from lines 271 to 277. Why would there be heterogeneity for one site but not for another? How does the data for Portugal relate to the data for North America where fourfold variations have been seen (a point which also arises in the text on line 305).
2. In the results it would be useful to include data about the stability of a single preparation of propolis stored over time in terms of the key active components, as well as parallel data for the stability of a blend, since these could respond differently. Stability of stored material over time is arguably just as important as stability in material that is made over time. The comment which is made in line 382 suggest that the authors do have such additional data on hand for the ethanolic extracts.
3. The section of text from lines 139 to 142 could be reworded better to explain the logic that is being presented. It is true that the more one aggregates together different preparations into a blend, the less variation that one should see, but from the way the comment is presented, maintaining consistent composition is in effect one benefit rather than two. If the authors can identify more clearly what the second benefit is, then they should do it in the text here. This also relates to the text on lines 290 to 292.
4. If there are references regarding the antioxidant potential of propolis preparations from parts of the world outside Europe, then they should be sited on line 189 to provide greater context for the point being made about European samples.
5. The discussion of the outlier result on line 248 would be stronger if the authors could allude to specific components present in that P14.EE preparation which could explain its unusual behaviour with bacteria versus yeasts.
6. In line 366-367, the authors make the claim about propolis stimulating the immune system, and appropriate references needed to be cited to support this point.
7. The paper would benefit from some minor English-language editing, by native English language speaker, to address multiple typographical errors.
8. As well, attention is necessary in the references to make sure that these follow the correct format for MDPI journals, especially the use of italics for the journal name. This problem recurs in the first 17 listed references. Later references have issues with capitalisation, e.g. reference 20. Each reference needs to be checked manually for proper format.
Author Response
Responses to Reviewer
The standardisation issues with phytochemicals are a well-known challenge and this paper makes an interesting and important contribution to addressing the challenge by using the well-known agent propolis. Hence the relevance of the study is quite high.
Reply: The authors totally agree that this is a major issue and thank the acknowledgement and encouraging comment of the reviewer.
A number of minor changes are needed to improve the overall quality of the paper.
Reply: The authors acknowledge the reviewer´s comments and tried to meet all the suggestions made.
- In the introduction it would be worthwhile pointing out some of the variations in quality and yield of propolis from different countries. The text does discuss productivity and yields but not quality in terms of the desired biological activities, hence it would be useful to include this aspect.
Reply: The authors thank the reviewer´s suggestion. Although literature reporting/reviewing chemical composition and therapeutic properties of propolis is relatively abundant, works regarding factors affecting propolis production arescarcer. Also, despite the multiple bioactivities of propolis and its increasingly use in medicine, cosmetics and food industries, the correlation between propolis quality and biological activity has not been investigated in enough depth and a quality standard (Portuguese or European) is still lacking. Such quality standard is obviously needed in order to protect consumers and (honest) producers, with researchers recommending that regulatory agencies should establish quality parameters for propolis of a certain country. Nevertheless, some factors have been linked to the improvement of propolis quality and quantity and this information as well as the respective references were added in the introduction, as suggested (lines 56 – 73).
It also provides greater context in terms of any claim that a particular location is a source for propolis that is "rather unique". This aspect should be explained more - making reference to factors that might explain such stable chemical composition (e.g. less climate change than in other regions over the same period of time). This same point arises in the text from lines 271 to 277. Why would there be heterogeneity for one site but not for another? How does the data for Portugal relate to the data for North America where fourfold variations have been seen (a point which also arises in the text on line 305).
Reply: The authors thank the reviewer´s pertinent point. The chemistry of propolis samples from different parts of the world has been extensively studied, and notably, was found to be characteristic or even unique to specific geographical zones, in many cases allowing to classify propolis in different types - like the poplar type, to which propolis from Portugal and North America belong. Yet, the chemical composition of propolis is largely dependent on the flora visited by the bees in the vicinity of their hives and, despite the characteristic phytochemical similarities of propolis from within the same geographical zone at the macroscale, region-specific uniqueness and variation in chemical composition have also been reported (Falcão et al., 2013 Journal of the American Oil Chemists' Society, 90(11), 1729-1741; Pereira et al., 2022 Food Control, 139, 109071). This phenomenon is most probably related to region-specific flora which is very different around Gerês and Pereiro apiaries, with the particularity of Gerês being located in a protected area of the Peneda do Gerês National Park where the surrounding vegetation (Calluna spp., Castanea sp., Rubus sp. and Quercus sp.) is maintained along the years. Also, as suggested by the referee, less climate change was noticed in Gerês than in Pereiro over the same period of time, which can influence the local vegetation, and Apis mellifera has not only preference for certain vegetation but also a limited vegetation as food source.
The apparent uniqueness of propolis from Gerês may be further explained by the apiculture practices (organic beekeepingcontrary to what happens in Pereiro) as well as the standardized way of propolis production and harvesting. Differently from Pereiro propolis, samples from Gerês are produced and collected from special grids, rather than being harvested by the usual method of scraping during the annual cleaning of the beehive, as in Pereiro.
It was precisely this inherent chemical variability between propolis samples from Pereiro and Gerês as well as the variability between samples collected in different years from Pereiro that encouraged the present study.
A better explanation regarding the uniqueness of propolis from Gerês was provided in the text (lines 307 – 312).
The flora in the vicinity of the beehives and its diversity, as well as climate variations may account to the variations mentioned for North America and Portugal but, in this case, it is also important to note that such differences could also be due to other factors like the time and the technique of harvest, the solvent and extraction procedures used and the protocols for performing chemical/biological tests.
- In the results it would be useful to include data about the stability of a single preparation of propolis stored over time in terms of the key active components, as well as parallel data for the stability of a blend, since these could respond differently. Stability of stored material over time is arguably just as important as stability in material that is made over time. The comment which is made in line 382 suggest that the authors do have such additional data on hand for the ethanolic extracts.
Reply: The authors thank the reviewer´s comment. In fact, we have some additional data regarding this issue and the manuscript was updated accordingly (lines 401-408). As mentioned in the manuscript, all the ethanol extracts used in our study were prepared in the years the propolis samples were collected, studied by then, and comparing with the results here reported no significant changes were observed regarding the bioactivities assessed. In a recent work we also showed that a propolis leftover (with more than a year storage) shows antimicrobial activity (Pereira et al., 2022 Food Control, 139, 109071) and we have recently prepared a G18.EE (ethanol extract of propolis from Gerês harvested in 2018) from a stored G18 sample that shows the same behavior as the G18. EE prepared in the same year of harvesting (2018; Oliveiraet al. 2022, Molecules 27(11), 3533). Furthermore, although collected and prepared in different years (2011, 2012, 2013 and 2014), G.EEs chemical profiles show a huge similarity regarding the type of phenolic compounds, variations being mostly quantitative (Freitas et al., 2019 Food Research International 119, 622-633), supporting the great resemblance between G.EEs bioactivities´ profiles. Recently too, we found that the phenolic profiles of P10.EE and P13.EE (ethanol extracts of propolis from Pereiro) are the same as the ones obtained in the respective harvesting years (2010 and 2013; work in progress). However, key active compounds of propolis from Gerês (as well as from Pereiro) remain to be clearly identified not allowing to examine samples stability from such perspective. But authors acknowledge the pertinent referee comment and will consider it to future work.
- The section of text from lines 139 to 142 could be reworded better to explain the logic that is being presented. It is true that the more one aggregates together different preparations into a blend, the less variation that one should see, but from the way the comment is presented, maintaining consistent composition is in effect one benefit rather than two. If the authors can identify more clearly what the second benefit is, then they should do it in the text here. This also relates to the text on lines 290 to 292.
Reply: We acknowledge the reviewer's comment and the text was reworded trying a better explanation (lines 162 - 165 and 325 - 327).
- If there are references regarding the antioxidant potential of propolis preparations from parts of the world outside Europe, then they should be sited on line 189 to provide greater context for the point being made about European samples.
Reply: We are grateful for the reviewer's suggestion. References regarding the antioxidant potential of propolis preparations from parts of the world outside Europe were added to that sentence (lines 211 – 216).
- The discussion of the outlier result on line 248 would be stronger if the authors could allude to specific components present in that P14.EE preparation which could explain its unusual behaviour with bacteria versus yeasts.
Reply: Authors acknowledge the very pertinent comment. A chromatographic analysis would be relevant to chemically characterize and compare the samples either among themselves or even with others from other sources. This is indeed a very interesting issue that could provide a clear-cut distinction between propolis compounds responsible for anti-yeast and anti-bacteria activities. However, we have just focused this work on exploiting propolis mixtures and its potential towards propolis standardization, valorization and profitability rather than performing an in-depth chemical analysis which would be a future goal given the obtained results and others obtained more recently.
- In line 366-367, the authors make the claim about propolis stimulating the immune system, and appropriate references needed to be cited to support this point.
Reply: (the original lines were 336-337) Appropriate references were added as suggested (actual lines 371 and 372).
- The paper would benefit from some minor English-language editing, by native English language speaker, to address multiple typographical errors.
Reply: The authors thank the reviewer´s advice. The paper was read by a native English language speaker and changed accordingly.
- As well, attention is necessary in the references to make sure that these follow the correct format for MDPI journals, especially the use of italics for the journal name. This problem recurs in the first 17 listed references. Later references have issues with capitalisation, e.g. reference 20. Each reference needs to be checked manually for proper format.
Reply: The authors thank the reviewer´s careful observation. Every reference was checked manually and changes were highlighted in yellow.
Reviewer 3 Report
Summary:
Peixoto and coworkers investigate the properties of mixtures of propolis extracts that were gathered at different times from the same location or from two different regions. Propolis extracts have potentially interesting antimicrobial and antioxidant activities, but there can be great variation from batch to batch. The key outcome of the study was that the mixtures maintained bioactivity and tended to average out to somewhat consistent values across mixtures.
Broad comments:
The authors test a good variety of sample mixtures, but the work has some concerning methodology issues that were not discussed. The authors also missed an opportunity to discuss what key descriptors of the extract best correlate with bioactivity.
1. If we are considering the chemical constituents of the extracts, we would not expect the amount of phenolic compounds or flavonoid derivatives to change upon mixing. If these actual species were quantitated, the TFC and TPC should always be an average of the two individual mixtures. This is not what was reported in the paper, and is likely due to the indirect assays used.
a. TPC - The Folin-Ciocalteu method really measures the total reducing capacity, not just the phenol content. More discussion is needed to explain the values obtained for the mixtures. If the phenolic content was truly being tested, the TPC of the mixtures should always be an average of the individual samples.
b. TFC – The TFC method used also has limitations. A discussion would be useful, for example see Journal of Food and Drug Analysis, 2002, 10, 178.
2. TFC/TPC don’t appear to correlate with antibacterial/antifungal activity, and antioxidant activity appears to have a negative correlation. A discussion of whether these characterization techniques are still useful would be helpful.
Specific comments:
Line 80 – How does mixing batches of propolis overcome low yields? The yield isn’t changing, you are just able to use mixtures without any negative issues?
Line 93 - Table 1 – the significant difference notation is difficult to follow. Same comment for Table 5.
Line 96 – “GAE” needs to be defined when it is first used in the text
Line 103 – “mg GAE/ g propolis extract mg GAE/ g extract” seems like a mistake. Were the units repeated? Or should it be “mg GAE/ g extract and mg QE/ g extract per g of extract“ ?
Line 106 – How can the TPC of the extracts be lower than the individual samples (mP(P11.EE+P13.EE) and mP(P13.EE+P15.EE))?
Line 130 – How can the TFC be higher than in either individual sample (i.e., for mP(P11.EE+P13.EE), mP(P11.EE+P14.EE), mP(P12.EE+P13.EE), mP(P13.EE+P14.EE), and mP(P11.EE-P13.EE))? The statement about complexation with aluminum(III) salts needs further discussion if it is the explanation. It’s currently unsatisfactory
Line 136/Table 2 – With the exception of mixtures containing P15.EE, it is unexpected that the EC50 of each mixture is always near the value of the more potent individual extract (within error). What’s going on? Why is P15.EE different?
Line 190 – The sentence “Figure 1 clearly show …” and line 194 “It is 194 clear that there is a correlation …” are repetitive.
Line 237 – Full names of bacterial strains are given in the caption, but they appear in the text first (so the full names should go there)
Line 486 – The modified agar dilution protocol likely used higher inocula than recommended, potentially leading to higher MIC values. See Nature Protocols, 2008, 3, 163 for a discussion.
Author Response
Response to Reviewer
Summary:
Peixoto and coworkers investigate the properties of mixtures of propolis extracts that were gathered at different times from the same location or from two different regions. Propolis extracts have potentially interesting antimicrobial and antioxidant activities, but there can be great variation from batch to batch. The key outcome of the study was that the mixtures maintained bioactivity and tended to average out to somewhat consistent values across mixtures.
Broad comments:
The authors test a good variety of sample mixtures, but the work has some concerning methodology issues that were not discussed. The authors also missed an opportunity to discuss what key descriptors of the extract best correlate with bioactivity.
Reply: The authors acknowledge the reviewer´s comments and tried to meet all the suggestions to improve the paper. We also would like to highlight that although literature reporting/reviewing chemical composition and multiple bioactivities of propolis is relatively abundant, the correlation between propolis quality and biological activity has not been investigated in enough depth and quality standards are still lacking. Such quality standards are obviously needed in order to protect consumers and producers and to allow propolis acceptance as an authenticated drug or health-promoting product in international food and healthcare markets, with researchers recommending that regulatory agencies should establish quality parameters for propolis of a certain country. However, propolis characteristics present several challenges to its standardisation and quality assessment and control. Until reliable and accurate analytical techniques are developed to be routinely and universally used, simple and fast characterization techniques (TPC, TFC, DPPH. Antimicrobial assays) are used and considered reliable for the purposes of most studies and for comparisons purposes.
- If we are considering the chemical constituents of the extracts, we would not expect the amount of phenolic compounds or flavonoid derivatives to change upon mixing. If these actual species were quantitated, the TFC and TPC should always be an average of the two individual mixtures. This is not what was reported in the paper, and is likely due to the indirect assays used.
Reply: The authors acknowledge the reviewer´s comments. In fact, if the assays simply quantified phenolic compounds or flavonoid derivatives, TFC and TPC of the mixtures would always be an average of the the individual components. These characterization techniques have indeed some limitations (see a. and b.) but are commonly used to provide an overview of the quality of propolis samples and generally accepted for such purpose. Particularly in this work, with several samples and comparison purposes, the chosen assays should be simple, fast, reproducible and reliable and the determination of TPC and TFC as well as the evaluation of the antioxidant and antimicrobial activities met those criteria. Nevertheless, some discussion about the mentioned methodologies and propolis quality issues were added to the manuscript (lines 58 – 73, 120 -128; 150 – 153).
- TPC - The Folin-Ciocalteu method really measures the total reducing capacity, not just the phenol content. More discussion is needed to explain the values obtained for the mixtures. If the phenolic content was truly being tested, the TPC of the mixtures should always be an average of the individual samples.
Reply: The authors acknowledge the reviewer's remark.
Nowadays, the Folin-Ciocalteu reagent is commercially available from many important commercial companies and this method is used extensively to quantify polyphenols in plant‐derived extracts (Lamuela‐Raventós 2018, Recent Trends and Applications, 107-115). As mentioned by the reviewer, the Folin-Ciocalteu method does not measure phenols, but the reducing activity that is assumed to be of phenols. And measuring the reducing capacity of the extracts in mixtures, for further expression as phenolic content in the blend, can result in values different than the average of the individual samples (Noreen et al., 2017. Asian Pacific Journal of Tropical Medicine, 10(8), 792-801). Providing an estimation of the phenolics, TPC can be affected by any compounds with reducing capacity and/or synergism/antagonism between reducing phenolics in complex mixtures. More discussion regarding this issue and adequate references were added to the manuscript (lines 120 -128).
Noteworthy, the Folin-Ciocalteu method is often criticized as giving higher values for polyphenols compared to the sum of the individual compounds as measured by HPLC (Mursu et al., 2008). However, the oligomers and polymers contain multiple phenolic groups and oxidation of them may produce products that are themselves reducing agents thus giving a greater Folin value. This sequence can potentially occur in vivo and thus the Folin measurement may be relevant and it is easier, quicker and less expensive to do the laboratory (Agbor et al., 2014).
- TFC – The TFC method used also has limitations. A discussion would be useful, for example see Journal of Food and Drug Analysis,2002, 10, 178.
Reply: The authors thank the referee comment and the literature suggestion. None of the current colorimetric methods used to measure TFC can detect all kinds of flavonoids, like the assay used in the present work, as not all flavonoids can stably form a complex with aluminum chloride (Chang et al. 2002). Considering the need of some parameters that would enable authors to compare propolis samples to each other, colorimetric assays, although not ideal methods due to their limitations, are convenient and generally accepted for routine analysis, providing an overview of the quality of the samples. Furthermore, AlCl3-based methodology is the most used and described assay to determine TFC. Nevertheless, a discussion on the limitations of the TFC methodology based on AlCl3 was added to the manuscript (lines 150 – 153).
- TFC/TPC don’t appear to correlate with antibacterial/antifungal activity, and antioxidant activity appears to have a negative correlation. A discussion of whether these characterization techniques are still useful would be helpful.
Reply: The authors acknowledge the referee observation and suggestion. The negative correlation between TFC/TPC and EC50 of radical scavenging activity is expected since flavonoids and polyphenols are regarded as the most active secondary metabolites of plants in terms of antioxidant activity.
Concerning the antibacterial/antifungal activity, the lack of correlation is expected since the antimicrobial molecular mechanisms against fungi and bacteria are quite different due to the different targets in prokaryotic and eukaryotic cells (also corroborated by the known mechanisms of activity and the chemical diversity of antibiotics and antifungals). In addition, although polyphenols and flavonoids might account for these activities, other compounds might be involved and synergisms between propolis compounds as well as between propolis and antibiotics/antifungals or between propolis and other natural products have been documented. Therefore, TPC and TFC are actually not useful to predict antimicrobial activity and in fact we only discussed this topic related to P14.EE with low TPC and TFC and high antibacterial activity (but not anti-yeast). So, due to the high complexity of the samples (individual extracts and mixtures) and the fact that antimicrobial metabolites are quite diverse, that discussion would not be useful.
TFC and TPC do not correlate with antibacterial/antifungal activity, not even with the antioxidant activity of propolis samples in several studies. However, as mentioned above and despite their limitations, all these characterization techniques are commonly used to evaluate the quality of propolis samples. Being fast, reliable, low cost and easy to perform in the laboratory, the methods allow to compare different propolis samples and to have an overview of its quality. Therefore, we think that the characterization techniques we used are reliable for the purposes of this study and they also allow to make comparisons with other studies.
Line 80 – How does mixing batches of propolis overcome low yields? The yield isn’t changing, you are just able to use mixtures without any negative issues?
Reply: The authors thank the reviewer´s pertinent questions. Our rationale was: as propolis yield (input) is low, the use of mixtures makes it possible to have larger amount of propolis extract for the market (output). But in fact, strictly speaking, the yield does not improve by mixing batches of propolis; instead, the amount of propolis available on the market increases by using and mixing propolis samples with different origins and age as well as propolis leftovers. Thus, we changed the sentence to make it clearer (lines 92 – 95).
Line 93 - Table 1 – the significant difference notation is difficult to follow. Same comment for Table 5.
Reply: The authors acknowledge and thank the reviewer´s comment. We tried to make such statistical notation more legible (using a semicolon and increasing the space between lower and capital letters in Tables 1, 2, 5 and 6) although keeping the same approach, as we have already used it in our previous work with mixtures from Gerês (Freitas et al2019). Still, if the referee has a better suggestion, we are happy to amend.
Line 96 – “GAE” needs to be defined when it is first used in the text
Reply: GAE as well as QE were defined the first time used in the text, as requested (lines 95-96).
Line 103 – “mg GAE/ g propolis extract mg GAE/ g extract” seems like a mistake. Were the units repeated? Or should it be “mg GAE/ g extract and mg QE/ g extract per g of extract“ ?
Reply: Authors acknowledge the careful review of the referee and the pertinent observation. The units were in fact repeated.
Line 106 – How can the TPC of the extracts be lower than the individual samples (mP(P11.EE+P13.EE) and mP(P13.EE+P15.EE))?
Reply: These results are truly intriguing. However, due to the chemical basis of the TPC estimation methodology (by reduction of Folin-Ciocalteu reagent), synergisms and/ or antagonisms might occur and influence the ability of polyphenols to react with Folin-Ciocalteu (please do see response to comment 1a too). The discussion of these results was included in the new version of the manuscript (lines 120 – 128).
Line 130 – How can the TFC be higher than in either individual sample (i.e., for mP(P11.EE+P13.EE), mP(P11.EE+P14.EE), mP(P12.EE+P13.EE), mP(P13.EE+P14.EE), and mP(P11.EE-P13.EE))? The statement about complexation with aluminum(III) salts needs further discussion if it is the explanation. It’s currently unsatisfactory
Reply: Following comment 1b (please do see response to this comment 1b) and these observations, we added a discussion regarding the different types of flavonoids detected by the method of AlCl3 reported by Chang et al (2002) (lines 150 – 153).
Line 136/Table 2 – With the exception of mixtures containing P15.EE, it is unexpected that the EC50 of each mixture is always near the value of the more potent individual extract (within error). What’s going on? Why is P15.EE different?
Reply: The reviewer is pointing out a very interesting feature of our study. The fact that mixtures of different extracts might increase the radical scavenging activity when compared with the individual extracts strongly suggests synergisms between samples. In our opinion this is not unexpected since synergisms (and antagonisms) have been widely reported in the literature. In addition, we also believe that the reasons for this are extremely complex because it involves interaction of compounds (2 or 3 or even more) from extracts of each mixture which are already very complex. The different behavior of mixtures with P15.EE might be a consequence of its unique chemical composition. An explanation regarding differences between propolis from Gerês and propolis from Pereiro was provided in the text (lines 307 – 312) and are related with the local flora, apiculture practices, propolis production and harvesting techniques. Seasonality and climate variations may further account to the differences between P.EEs.
Line 190 – The sentence “Figure 1 clearly show …” and line 194 “It is 194 clear that there is a correlation …” are repetitive.
Reply: Authors acknowledge the careful review of the referee and the pertinent observation. The text was checked, the repetitive part deleted and the first sentence rewritten (lines 217 - 219).
Line 237 – Full names of bacterial strains are given in the caption, but they appear in the text first (so the full names should go there)
Reply: Authors acknowledge once again the pertinent comment of the referee. Full names of bacterial strains were given when first appeared.
Line 486 – The modified agar dilution protocol likely used higher inocula than recommended, potentially leading to higher MIC values. See Nature Protocols, 2008, 3, 163 for a discussion.
Reply: We thank the reviewer for pointing this part of the method for assessing MIC as well as the literature suggestion. In fact, we used a considerable higher inoculum than recommended by Wiegand et al (2008; Wiegand et al. 2008. Agar and broth dilution methods to determine the minimal inhibitory concentration (MIC) of antimicrobial substances. Nature protocols, 3(2), 163-175), as we usually do in our studies, and the main reasons for doing that were to avoid false positives (from low inocula) and to provide more robust results (lower inocula might be prone to variations in results in replicas). According to the same authors, higher inocula influence MIC especially in the case of bacteria that segregate enzymes that degrade antimicrobial agents. In our opinion, such situation is unlikely to occur in our experiments since we are testing complex extracts and their mixtures so, the possibility of any tested bacterium to segregate enzymes to all of them can be disregarded and even if such enzymes would be produced, they would only degrade a very low proportion of the antibacterial compounds of the extracts and mixtures (which does not seem the case as mixtures showed to maintain/improve antimicrobial properties).
Round 2
Reviewer 2 Report
All of my concerns have been addressed by the changes made